# Fuzzy Color Aura Matrices for Texture Image Segmentation

**DOI:** 10.3390/jimaging8090244

**Published:** 2022-09-08

**Authors:** Zohra Haliche, Kamal Hammouche, Olivier Losson, Ludovic Macaire

**Affiliations:** 1Laboratoire Vision Artificielle et Automatique des Systèmes, Université Mouloud Mammeri, Tizi-Ouzou 15000, Algeria; 2Univ. Lille, CNRS, Centrale Lille, UMR 9189 CRIStAL, F-59000 Lille, France

**Keywords:** color texture segmentation, aura matrices, fuzzy color aura matrix, SLIC superpixel, regional feature

## Abstract

Fuzzy gray-level aura matrices have been developed from fuzzy set theory and the aura concept to characterize texture images. They have proven to be powerful descriptors for color texture classification. However, using them for color texture segmentation is difficult because of their high memory and computation requirements. To overcome this problem, we propose to extend fuzzy gray-level aura matrices to fuzzy color aura matrices, which would allow us to apply them to color texture image segmentation. Unlike the marginal approach that requires one fuzzy gray-level aura matrix for each color channel, a single fuzzy color aura matrix is required to locally characterize the interactions between colors of neighboring pixels. Furthermore, all works about fuzzy gray-level aura matrices consider the same neighborhood function for each site. Another contribution of this paper is to define an adaptive neighborhood function based on information about neighboring sites provided by a pre-segmentation method. For this purpose, we propose a modified simple linear iterative clustering algorithm that incorporates a regional feature in order to partition the image into superpixels. All in all, the proposed color texture image segmentation boils down to a superpixel classification using a simple supervised classifier, each superpixel being characterized by a fuzzy color aura matrix. Experimental results on the Prague texture segmentation benchmark show that our method outperforms the classical state-of-the-art supervised segmentation methods and is similar to recent methods based on deep learning.

## 1. Introduction

Color texture segmentation consists of partitioning an image into homogeneous regions with respect to color and texture properties. It is involved in various fields, such as medical image analysis [1], remote sensing [2], synthetic aperture radar [3], and fruit detection [4]. Although a wide variety of techniques have been developed, color texture segmentation remains an open and challenging problem due to the high variability of textures, combined with the great diversity of colors. Most approaches characterize each pixel by a set of texture and color features, and use them to assign one color texture class to each pixel thanks to classification algorithms [5,6]. Section 1.1 and Section 1.2 discuss the state of the art of color texture features and classification algorithms used for color texture segmentation.

### 1.1. Color Texture Features

The characterization of color textures is a fundamental problem in computer vision, where a texture is generally described by some visual cues represented by statistical structures, hereafter called texture features. Pixel colors are often represented by the values or by statistic measures of color components (R, G, B) or by those derived from color spaces such as HSV, L${}^{*}$u${}^{*}$v${}^{*}$, or L${}^{*}$a${}^{*}$b${}^{*}$ [7]. Various texture features are designed to characterize texture appearance. Widely used ones are based on Gabor filters, wavelet transform, Markov random field model, local binary patterns, and co-occurrence matrices [8,9,10].

Color texture classification/segmentation techniques can be broadly classified into two approaches according to how they combine color texture and color features [5,8]. In the first approach, color and texture features are computed separately and are then combined by the clustering or classification process. For example, in [11], L${}^{*}$, u${}^{*}$, and v${}^{*}$ components were used as color features and the output of Gabor filters as texture features. In [12], the authors used the mean, standard deviation, and skewness of each color channel, H, S, and V, as color features, and the co-occurrence matrices computed on the intensity image were considered as texture features.

In the second approach, color and texture features are assumed to be mutually dependent. With this approach, marginal, opponent, and vector strategies have been developed to compute color texture features [5,8]. The marginal strategy assumes that texture can be separately described within each color channel, where pixels are characterized by only one color component. Texture features designed for gray-level images are then computed for each color channel and aggregated into a global feature representation. In [13], for instance, texture features were extracted from wavelet transform and co-occurrence matrices for each color channel. In [14], Leung–Malik filter banks were applied on each color channel for color texture feature extraction. The opponent strategy extends texture feature extraction to color images thanks to both within- and between-channel analyses. The within-channel part consists of computing features from each color channel separately (as in the marginal strategy), whereas between-channel (opponent) features are obtained by processing pairs of color channels jointly. This strategy was adopted by Panjwani et al. [15], who used Markov random field models to characterize a texture in terms of spatial interaction within each color channel and interaction between different color channels. The vector strategy takes advantage of color vector information. It makes it possible to analyze relationships between colors of neighboring pixels, as in the case of gray-level images. Examples of this strategy can be found in [9,16] where local binary patterns were computed from a color image. It should be noted that the vector strategy is more suitable to characterizing color texture because it is less memory-consuming and fully takes correlation between colors of neighboring pixels into account.

### 1.2. Color Texture Image Segmentation by Pixel Classification

Color texture segmentation methods can also be categorized into three ways, depending on whether the pixel classification is performed in a supervised, unsupervised, or semi-supervised manner [1,11,17]. In supervised segmentation, prior knowledge about the training samples and their class labels is needed to classify the input image pixels. Some classical supervised classifiers are used for texture segmentation [1,4,17]. Among them are the *K*-nearest neighbor (*K*NN) [18] and the Bayesian [19] classifiers, the support vector machine (SVM) [20], random forest [12], Markov random field [11], and neural networks [14]. Supervised color texture segmentation based on deep learning has been developed in the last decade [21,22,23]. These efficient methods generally use convolutional neural networks (CNN), such as the U-Net [24], deep visual model (DA) [25], pyramid scene parsing network (PSP-Net) [26], supervised fully convolutional network for texture (FCNT) [21], and empirical wavelet transform-based fully convolutional network for texture (EWT-FCNT) [23].

Unsupervised segmentation does not require prior knowledge and discovers different classes by clustering pixels from their features only [5,6]. For example, the popular *K*-means clustering algorithm is mainly employed to perform the classification of pixels [6,27].

Other color texture segmentation methods rely on graph cut techniques [28], mean shift clustering [29], estimation–maximization [1], the random Markov model [11], dictionary learning and sparse representation-based classification (DLSRC) [13], and spectral clustering [3].

Semi-supervised segmentation is suitable where only partial prior knowledge about the training samples and their class labels is available. For instance, in [17], the constrained spectral clustering algorithm was applied for semi-supervised classification of pixels characterized by color texture features.

Supervised segmentation methods, especially deep learning based methods, provide better results than unsupervised and semi-supervised approaches thanks to prior information [17]. It should also be noted that some of the above methods perform segmentation at the pixel level and others at the "superpixel" level. This is the case for the methods discussed in [1,3]. Recall that superpixels group neighboring pixels based on their spatial and color similarity, and are used as samples in order to speed up color texture segmentation.

### 1.3. Fuzzy Color Texture Features

All aforementioned methods assume that images are crisp and free from vagueness. However, in practice, color images carry more or less fuzziness because of the imprecision inherent to the discretization of both the spatial domain (sampling) and the color component levels (quantization). Therefore, boundaries separating the various image regions are not precisely defined, and pixel levels are imprecise measures of the reflectance of surfaces observed by the camera. Furthermore, the assumption that texture images are mainly represented by spatial repetitions of a pattern may not be valid any longer. Texture analysis techniques based on fuzzy concepts have then been proposed in order to take this imprecision into account.

For instance, fuzzy histograms [30], fuzzy local binary patterns [31], and local fuzzy patterns [32] are extracted from gray-level texture images. Fuzzy gray-level co-occurrence matrices (FGLCMs) are also proposed to characterize spatial interactions between gray-levels of neighboring pixels [33,34,35,36,37]. However, characterizing a color texture by at least three FGLCMs (one by channel) is memory expensive. To overcome this drawback, Ledoux et al. [38] proposed to extend FGLCMs to color images by defining fuzzy color sets. A color image is then represented by one single fuzzy color co-occurrence matrix (FCCM) that characterizes the local interactions between colors of neighboring pixels. Moreover, FGLCMs and FCCMs only consider spatially-invariant neighborhoods. Hammouche et al. [39] showed that adaptive neighborhoods are useful for texture analysis and provided an elegant formalism to deal with spatially-variant neighborhoods thanks to the aura concept.

In the framework based on the aura set concept, Elfadel and Picard [40] proposed a generalization of GLCMs called gray-level aura matrices (GLAMs). A GLAM quantifies the presence of a set of pixels with a specified level in the neighborhood of another set of pixels having another level. The amount of neighboring pixels with the specified level is quantified by means of the aura measure. GLAMs are used for texture representation and synthesis [41,42], image retrieval [43,44], classification [45,46,47,48,49,50], and segmentation [51,52,53]. A generalization of GLAMs to the fuzzy framework has been proposed by Hammouche et al. [39]. Representing each color channel by a fuzzy GLAM (FGLAM) outperforms the FGLCM representation for texture classification. Recently, FGLAMs have been used to improve the accuracy of wood species classification [54]. However, as for FGLCMs, the computation of FGLAMs for each pixel is memory and time expensive, which makes their use for color image segmentation a challenge.

To circumvent these constraints, we adopted a vector strategy and propose to extend FGLAMs to fuzzy color aura matrices (FCAMs). An FCAM makes it possible to locally characterize the interactions between colors of neighboring pixels. While one FGLAM must be computed for each color channel, a single low-dimensionnal FCAM is required to describe the color texture.

### 1.4. FCAM for Image Segmentation by Superpixel Classification

In this study, we applied FCAM to color texture image segmentation. As FCAM is based on a locally-adaptive neighborhood, it can characterize the texture represented by small connected pixel subsets with different shapes, i.e., by superpixels. We use the simple linear iterative clustering (SLIC) scheme [55] to generate superpixels from a color image. These are then classified using a simple supervised classifier to segment the color texture image.

The remainder of this paper is organized as follows. In Section 2, we first give an overview of basic SLIC algorithm; then we propose a modified version that incorporates the regional information. Section 3 introduces some definitions of the fuzzy color aura concept, and explains how to characterize the color texture of each superpixel by an FCAM. In Section 4, we first present the datasets used in the experiments; then we give details about the proposed color texture segmentation method based on FCAMs. Next, we assess the regional SLIC algorithm and discuss its parameter settings. Last, we compare the segmentation results achieved by our supervised segmentation approach with those obtained by fuzzy texture features and by several other state-of-the-art color texture segmentation methods. Concluding remarks about the contribution of this paper are given in Section 5.

## 2. Superpixel

The proposed color texture image segmentation is based on the classification of superpixels. A superpixel is a compact set of connected sites with similar properties. It is usually adopted to replace the pixel grid in an image in order to reduce the computational burden of subsequent processing. Superpixels are generated by over-segmentation algorithms using color, spatial, and/or texture information. Among superpixel generation algorithms, simple linear iterative clustering (SLIC) is widely used due to its simplicity, speed, and ability to adhere to image boundaries. In this section, we briefly give an overview of SLIC algorithm used to generate superpixels. Then, we propose a modified version that incorporates the regional information.

### 2.1. Basic SLIC

The SLIC algorithm [55] generates a desired number of regular superpixels, with la ow computational overhead, by clustering sites based on their spatial and color features. Let I be an RGB image defined on a lattice S, such that each site s∈S is characterized by three color components: $I\left(s\right)=(I^{R}\left(s\right)$, $I^{G}\left(s\right)$, $I^{B}{\left(s\right))}^{⊺}$. The RGB color components are transformed into the $La^{*}b^{*}$ color space, so that SLIC represents each site $s(x_{s},y_{s})$ by a five-dimensional feature vector: $(I^{L}\left(s\right)$, $I^{a}\left(s\right)$, $I^{b}\left(s\right),x_{s},y_{s}{)}^{⊺}$. SLIC follows a *k*-means clustering strategy but searches the nearest cluster center according to the distance $D\left(s,s^{\prime }\right)=\sqrt{d_{c}^{2}\left(s,s^{\prime }\right)+m^{2}\cdot d_{s}^{2}\left(s,s^{\prime }\right)/S^{2}}$ between two sites s and $s^{\prime }$, where $d_{c}^{2}\left(s,s^{\prime }\right)={\left(I^{L}\left(s\right)-I^{L}\left(s^{\prime }\right)\right)}^{2}+{\left(I^{a}\left(s\right)-I^{a}\left(s^{\prime }\right)\right)}^{2}+{\left(I^{b}\left(s\right)-I^{b}\left(s^{\prime }\right)\right)}^{2}$ and $d_{s}^{2}\left(s,s^{\prime }\right)={\left(x_{s}-x_{s^{\prime }}\right)}^{2}+{\left(y_{s}-y_{s^{\prime }}\right)}^{2}$ measure the color and spatial proximity. The compactness parameter *m* is set to its default value of 1, and the maximum spatial distance within a cluster is defined as the sampling step $S=\sqrt{\left|S\right|/P}$, where $\left|S\right|$ is the total number of sites in the image and *P* is the number of superpixels. Figure 1b shows the results of the segmentation achieved by SLIC on the color image of Figure 1a. It should be emphasized that SLIC takes a post-processing step to enforce connectivity by merging small isolated superpixels with nearby larger ones. Therefore, the actual number of superpixels produced by SLIC can be slightly lower than the desired number of superpixels.

### 2.2. Regional SLIC

The basic SLIC algorithm achieves good pre-segmentation on color images, but may fail to find boundaries of textures. Indeed, it successfully detects homogeneous regions but somewhat fails to separate areas with different textures (see Figure 1a and arrows on Figure 1b). To improve this pre-segmentation, we propose a modified version of the algorithm, referred to as regional SLIC, that takes color, spatial, and regional information into account. From the basic SLIC pre-segmentation result, we compute the regional feature $f_{R}$ of each superpixel R as [56]:(1)$$f_{R}=\left[p_{R}-p_{R}\cdot log\left(p_{R}\right)\right]+\frac{1}{|N_{R}|}\sum_{R^{\prime }\in N_{R}}\left[p_{R^{\prime }}-p_{R^{\prime }}\cdot log\left(p_{R^{\prime }}\right)\right],$$
where $p_{R}=\left|R\right|/\left|S\right|$ is the area ratio of R to the whole image, and $N_{R}$ is the set of superpixels that are adjacent to R. The regional feature $f_{R}$ is a sum of two terms: the first one reflects the size of the superpixel R and monotonously increases with its size (because $p_{R}\in ]0,1[$), and the second one reflects the superpixel context by taking the influence of adjacent superpixels into account. A small superpixel surrounded by small superpixels provides a low value of $f_{R}$, and conversely, if both R and adjacent superpixels are large, $f_{R}$ is high (see Figure 1c). At each site s of superpixel R, we replace the luminance component $I^{L}\left(s\right)$ by the regional feature $f_{R}$ and we apply the SLIC algorithm again with $(f_{R}$, $I^{a}\left(s\right)$, $I^{b}\left(s\right),x_{s},y_{s}{)}^{⊺}$ as the feature vector (see Algorithm 1).

Thanks to this regional SLIC method, the lattice S is partitioned into *P* superpixels ${\left\{P^{p}\right\}}_{p=1}^{P}$, i.e., ${⋃}_{p=1}^{P}P^{p}=S$ and $P^{p}\cap P^{p^{\prime }}=\emptyset$, for $p\neq p^{\prime }$. A superpixel $P^{p}$ is defined as a set of connected sites, i.e., any two sites s and $s^{\prime }$ in $P^{p}$ are connected by at least one path composed of sites in $P^{p}$. Figure 1d shows the final superpixels extracted from the color texture image of Figure 1a. Note (see arrows) that areas with different textures are better delineated with the regional SLIC than with its basic version, despite the regional feature describing the superpixel contextual information in the pre-segmented image and not being able to be considered as a texture feature.
**Algorithm 1:**Regional SLIC.**Input:** RGB image I1.Convert I from RGB to $La^{*}b^{*}$ color space;2.Apply SLIC to S where each site s is characterized by $(I^{L}\left(s\right)$, $I^{a}\left(s\right)$, $I^{b}\left(s\right),x_{s},y_{s}{)}^{⊺}$;3.Extract the regional feature $f_{R}$ from each superpixel R of the SLIC map using Equation (Equation 1);4.Apply SLIC to S where each site s is characterized by $(f_{R}$, $I^{a}\left(s\right)$, $I^{b}\left(s\right),x_{s},y_{s}{)}^{⊺}$.**Output:** Partition of I into superpixels ${\left\{P^{p}\right\}}_{p=1}^{P}$


## 3. Fuzzy Color Aura

In this section, we extend the aura concept introduced by Elfadel and Picard for gray-level images [40] to color images. We explain how it is used to characterize a superpixel thanks to a matrix of aura cardinals. As this matrix is huge when all RGB colors are considered, we propose to extend the color aura concept to fuzzy colors whose number is reduced.

### 3.1. Color Aura Set

Let I be an RGB image defined on a lattice S, such that each site s∈S is characterized by its color I(s). For a given color x∈RGB, we define the set of sites with this color as $S_{x}=\left\{s\in S,I\left(s\right)=x\right\}$. Then, ${\left\{S_{x}\right\}}_{x\in \mathrm{RGB}}$ is a partition of S—i.e., ${⋃}_{x\in \mathrm{RGB}}S_{x}=S$ and $S_{x}\cap S_{x^{\prime }}=\emptyset$ for $x\neq x^{\prime }$.

Given two colors $\left(x,x^{\prime }\right)\in {\mathrm{RGB}}^{2}$, we define the color aura set $A_{S_{x^{\prime }}}\left(S_{x}\right)$ of $S_{x}$ with respect to $S_{x^{\prime }}$ as:(2)$$A_{S_{x^{\prime }}}\left(S_{x}\right)=\underset{s\in S}{⋃}\left(N_{S_{x}}^{s}\cap S_{x^{\prime }}\right),$$
where $N_{S_{x}}^{s}$ is the neighboring site set of each site $s\in S_{x}$:(3)$$N_{S_{x}}^{s}=\left\{r\in S,\left(s\in S_{x}\right)\wedge \left(r\in N^{s}\right)\right\}.$$

$N^{s}\overset{\cdot }{=}\left\{{r\in S,∥r-s∥}_{\infty }\leq d\right\}$ is the neighborhood of a site s. In this paper, d=1 such that $N^{s}$ is only composed of the eight closest sites from s.

$A_{S_{x^{\prime }}}\left(S_{x}\right)$ is the subset of $S_{x^{\prime }}$ composed of the sites that are present in the neighborhood of those of $S_{x}$. It provides an interpretation of the presence of $S_{x^{\prime }}$ in the neighborhood of $S_{x}$. The color aura set is a generalization of the gray-level aura set introduced by Elfadel and Picard [40] to the color case.

Figure 2a shows a color image, defined on a lattice S of 7×7 pixels, and three color sets, $S_{R},S_{G}$, and $S_{B}$. For the neighborhood $N^{s}$, the aura set $A_{S_{G}}\left(S_{B}\right)$ of $S_{B}$ with respect to $S_{G}$ is composed of the green sites marked as circles. Note that it differs from the aura set $A_{S_{B}}\left(S_{G}\right)$ of $S_{G}$ with respect to $S_{B}$, composed of the blue sites marked as diamonds.

### 3.2. Color Aura Set in a Superpixel

Rather than building an aura set from all image sites, we focus on a superpixel $P^{p}$ and define the color aura set $A_{S_{x^{\prime }}}^{p}\left(S_{x}\right)$ of $S_{x}$ with respect to $S_{x^{\prime }}$ for the superpixel $P^{p}$ as:(4)$$A_{S_{x^{\prime }}}^{p}\left(S_{x}\right)=\underset{s\in S}{⋃}\left(N_{S_{x}}^{p,s}\cap S_{x}^{\prime }\right),$$
where $N_{S_{x}}^{p,s}$ is the neighboring site set of each site $s\in S_{x}$ within the superpixel $P^{p}$:(5)$$N_{S_{x}}^{p,s}=\left\{r\in S,\left(s\in S_{x}\cap P^{p}\right)\wedge \left(r\in N^{s}\cap P^{p}\right)\right\}.$$

The image of Figure 2a contains two textures, one represented by red and green vertical stripes on the left three columns, and another represented by red and blue horizontal stripes on the right three columns. These two textures are separated by the fourth column composed of red, green, and blue sites. Figure 2b shows the partitioning of this image into two superpixels delimited by a vertical black line. $P^{1}$ covers the four left columns and $P^{2}$ the three right ones. The aura sets $A_{S_{G}}^{1}\left(S_{B}\right)$ and $A_{S_{B}}^{1}\left(S_{G}\right)$ are not empty, since $P^{1}$ contains neighboring green and blue sites, but $A_{S_{G}}^{2}\left(S_{B}\right)$ and $A_{S_{B}}^{2}\left(S_{G}\right)$ are empty because $P^{2}$ contains no green site. Figure 2c shows another partition where $P^{1}$ covers the three left columns and $P^{2}$ the four right ones. In that case, $A_{S_{G}}^{1}\left(S_{B}\right)$ and $A_{S_{B}}^{1}\left(S_{G}\right)$ are empty because $P^{1}$ contains no blue site. This example illustrates that color aura sets depend on superpixel edges. Indeed, along the fourth column that separates the two textures, only one green site (on first row) and one blue site (second row) belong to the aura sets of superpixels in both Figure 2b,c.

### 3.3. Color Aura Cardinal

The aura measure was introduced by Elfadel and Picard [40] to characterize an aura set by a number that expresses the amount of mixing between neighboring site sets. We use here a simpler measure [39] and quantify the color aura set of a color site set $S_{x}$ with respect to another color site set $S_{x^{\prime }}$ within the superpixel $P^{p}$ thanks to its cardinal defined as:(6)$$m^{p}\left(x,x^{\prime }\right)=\left|A_{S_{x^{\prime }}}^{p}\left(S_{x}\right)\right|.$$

The aura cardinal measures for all the possible pairs of color sets, $\left(S_{x},S_{x^{\prime }}\right)$, $\left(x,x^{\prime }\right)\in {\mathrm{RGB}}^{2}$, are gathered in a matrix. This color aura cardinal matrix $\left[m^{p}\left(x,x^{\prime }\right)\right]$ can be then considered as the texture feature of the superpixel $P^{p}$. However, when color components of the image are defined on 256 levels, as is classically the case, the number of possible colors reaches $256^{3}$ and $m^{p}$ would be of size $256^{3}\times 256^{3}$. As such, a huge memory requirement is needed in practice, so we propose to decrease the number of analyzed colors by introducing fuzzy colors.

### 3.4. Fuzzy Color

To form a small subset C of *C* colors among the $256^{3}$ possible ones, we use the uniform quantization technique for its simplicity of implementation. For each color component k∈{R,G,B}, the full level range [[0,255]] is divided into $C^{k}$ disjoint intervals $\left[\;\left[0,L^{k}-1\right]\;\right]$, $\left[\;\left[L^{k},2L^{k}-1\right]\;\right],\cdots ,\left[\;\left[\left(C^{k}-1\right)L^{k},255\right]\;\right]$, of respective width $L^{k}\overset{\cdot }{=}256/C^{k}$. The $C^{k}$ centers $\left\{\left\lfloor \left(L^{k}-1\right)/2\right\rfloor ,\left\lfloor \left(3L^{k}-1\right)/2\right\rfloor ,\cdots ,\left\lfloor 256-\left(L^{k}+1\right)/2\right\rfloor \right\}$ of the intervals define the *k*-th component of the colors in C. The numbers $C^{R}$, $C^{G}$, and $C^{B}$ are chosen so that the number of colors $C=C^{R}\cdot C^{G}\cdot C^{B}$ is much lower than $256^{3}$.

In the fuzzy framework [38], a fuzzy color $\tilde{c}$ is characterized by its membership function $\mu_{\tilde{c}}:\mathrm{RGB}⟶\left[0,1\right]$. The membership degree $\mu_{\tilde{c}}\left(x\right)$ of any color x∈RGB is defined using its infinity norm or Euclidean distance to the crisp counterpart c∈C of $\tilde{c}$, thanks to either:the crisp membership function:
(7)$$\mu_{\tilde{c}}\left(x\right)=\left\{\begin{array}{cc} 1 & {\mathrm{if}\|x-c\|}_{\infty }\leq \left\lfloor \frac{L^{k}}{2}\right\rfloor ,\\ 0 & \mathrm{otherwise}, \end{array}\right.$$the symmetrical Gaussian function:
(8)$$\mu_{\tilde{c}}\left(x\right)=\exp\left(-\frac{{\left\|x-c\right\|}_{2}^{2}}{2\alpha^{2}}\right)\;,$$the triangular function:
(9)$$\mu_{\tilde{c}}\left(x\right)=\max\left(1-\frac{{\left\|x-c\right\|}_{2}}{\beta },0\right)\;,$$or the fuzzy *C*-means (FCM) membership function:
(10)$$\mu_{\tilde{c}}\left(x\right)=\frac{1}{\sum_{c^{\prime }\in C}{\left(\frac{{\left\|x-c\right\|}_{2}}{\|x-c^{\prime }{\|}_{2}}\right)}^{\frac{2}{\zeta -1}}}\;.$$

Here, α and β are real positive constants used to control the span of the fuzzy color, and ζ is any real number greater than 1. In this paper, we set $\beta =(L^{R},L^{G},L^{B})$, $\alpha =\beta /\sqrt{2\ln(2)}$, and ζ=2. The parameters α and β are chosen to ensure that $\mu_{\tilde{c}}\left(x\right)=0.5$ at the bounds of each color domain of width ${\left(L^{R},L^{G},L^{B}\right)}^{⊺}$ centered at c. Figure 3 shows the shapes of the four membership functions computed at c=(192,64,64), for plane (R,G).

### 3.5. Fuzzy Color Aura Set in a Superpixel

A fuzzy color site set $S_{c}$, c∈C, is defined by its membership function $\mu_{S_{c}}$. The membership degree $\mu_{S_{c}}\left(s\right)$ of each site s∈S to $S_{c}$ is the membership degree $\mu_{\tilde{c}}\left(I\left(s\right)\right)$ of its color I(s)∈RGB to the fuzzy color $\tilde{c}$ [38]:(11)$$\mu_{S_{c}}\left(s\right)=\mu_{\tilde{c}}\left(I\left(s\right)\right).$$

From there, we define the fuzzy color aura set $A_{S_{c^{\prime }}}^{p}\left(S_{c}\right)$ of $S_{c}$ with respect to $S_{c^{\prime }}$, for any color pair $\left(c,c^{\prime }\right)\in C^{2}$, as the fuzzy site set with the following membership degree at each site r∈S:(12)$$\mu_{A_{S_{c^{\prime }}}^{p}\left(S_{c}\right)}\left(r\right)=\min\left\{\underset{s\in S}{\sup}\left[\min\left(\mu_{S_{c}}\left(s\right),n^{p}\left(r,s\right)\right)\right],\mu_{S_{c^{\prime }}}\left(r\right)\right\}.$$

The neighborhood function $n^{p}\left(r,s\right)$ expresses the membership degree of r to the neighborhood of s within the superpixel $P^{p}$, p∈[[1,P]]. This function may have any support size and shape, and may take any real value between 0 and 1. In the simplest binary case, we design it as:(13)$$n^{p}\left(r,s\right)=\left\{\begin{array}{cc} 1 & \mathrm{if}\;\left(s\in P^{p}\right)\wedge \left(r\in N^{s}\cap P^{p}\right),\\ 0 & \mathrm{otherwise}. \end{array}\right.$$

As a justification of Equation (Equation 12), we consider the fuzzy color aura set by analogy with the crisp case [39]. The crisp set union operator ⋃ of Equation (Equation 4) is transcribed by the fuzzy operator sup to get the membership degree of the fuzzy color aura set $S_{c}$ with respect to $S_{c^{\prime }}$ at each site r∈S:(14)$$\mu_{A_{S_{c^{\prime }}}^{p}\left(S_{c}\right)}\left(r\right)=\underset{s\in S}{\sup}\left(\mu_{N_{S_{c}}^{p,s}\cap S_{c^{\prime }}}\left(r\right)\right).$$

The crisp set intersection operator ∩ is transcribed by the fuzzy operator min, such that the fuzzy counterpart of the color site set $N_{S_{x}}^{p,s}\cap S_{x^{\prime }}$ of Equation (Equation 4) is defined by its membership function given by:(15)$$\mu_{N_{S_{c}}^{p,s}\cap S_{c^{\prime }}}\left(r\right)=\min\left[\mu_{N_{S_{c}}^{p,s}}\left(r\right),\mu_{S_{c^{\prime }}}\left(r\right)\right].$$

The fuzzy counterpart of $N_{S_{x}}^{p,s}$ (see Equation (Equation 5)) is defined for the fuzzy color site set $S_{c}$, c∈C, by the following membership degree at each site r∈S:(16)$$\mu_{N_{S_{c}}^{p,s}}\left(r\right)=\min\left(\mu_{S_{c}}\left(s\right),n^{p}\left(r,s\right)\right).$$

Plugging Equation (Equation 16) into (Equation 15) and the result into (Equation 14) provides the definition (Equation 12) of the fuzzy aura set after swapping the first two operators.

### 3.6. Fuzzy Color Aura Cardinal

The fuzzy color aura cardinal of $S_{c}$ with respect to $S_{c^{\prime }}$ within $P^{p}$ is exactly defined as in the crisp case (see Equation (Equation 6)) and directly follows from Equation (Equation 12):(17)$${\tilde{m}}^{p}\left(c,c^{\prime }\right)=\sum_{r\in P^{p}}\mu_{A_{S_{c^{\prime }}}^{p}\left(S_{c}\right)}\left(r\right)=\sum_{r\in P^{p}}\min\left\{\underset{s\in P^{p}}{\sup}\left[\min\left(\mu_{S_{c}}\left(s\right),n^{p}\left(r,s\right)\right)\right],\mu_{S_{c}^{\prime }}\left(r\right)\right\}.$$

Following the crisp scheme, we define a fuzzy color aura matrix (FCAM) as the collection of all fuzzy color aura cardinals of a superpixel for the $C^{2}$ color pairs $\left(c,c^{\prime }\right)\in C^{2}$. Note that to define the FCAM, we only consider a few of the $256^{3}$ colors on which the RGB image is defined—namely, the set C of *C* colors such that $C\ll 256^{3}$. This provides a compact FCAM of small size C×C that is a suitable color texture descriptor of a superpixel for image segmentation purpose.

To make superpixels of different sizes comparable, their FCAMs are normalized element-wise to sum up to one:(18)$${\bar{m}}^{p}\left(c,c^{\prime }\right)=\frac{{\tilde{m}}^{p}\left(c,c^{\prime }\right)}{\sum_{\left(y,y^{\prime }\right)\in C^{2}}{\tilde{m}}^{p}\left(y,y^{\prime }\right)}.$$

Finally, each superpixel $P^{p}$ of an image I is characterized by $C^{2}$ features that are the elements of its normalized FCAM ${\bar{m}}^{p}$.

## 4. Experiments

In this section, we first present the experimental dataset and explain how FCAM features are used to segment its color texture images. To evaluate the performance of the proposed approach, we then assess the regional SLIC algorithm, discuss parameter settings, and study the relevance of FCAM features. We finally compare the segmentation results achieved by our supervised segmentation approach with those obtained by several state-of-the-art color texture segmentation methods.

### 4.1. Experimental Setup

#### 4.1.1. Dataset

As the experimental dataset, we used the challenging Prague texture segmentation benchmark [57]. It contains 20 color texture mosaics to be segmented (input test images), some of which are shown in Figure 4 (top) with their corresponding segmentation ground truth images. Each of these images represents from 3 to 12 classes and has been synthetically generated from the same number of original color texture images. The original Prague dataset is composed of 89 images (one image per texture class) grouped into 10 categories of natural and artificial textures. Figure 4 (bottom) shows some of these images that form the training dataset. All images of Prague database are of size 512×512 pixels.

#### 4.1.2. Color Texture Image Segmentation Based on FCAMs

Our method of color texture image segmentation based on FCAMs is a supervised superpixel classification procedure. Each test image I represents $K_{I}$ classes whose pixels should be retrieved by the segmentation. To this end, we used a simple artificial neural network known as the extreme learning machine (ELM) that comes with a very fast learning algorithm [58]. It consists of three fully-connected neuron layers: an input layer of $C^{2}$ neurons that receives FCAM elements, a single hidden layer, and an output layer with $K_{I}$ neurons. The number of neurons in the hidden layer was empirically set to $100\cdot K_{I}$. The initial weights of hidden neurons were set to random values, and the output weights were determined according to a least-square solution [58].

The proposed color texture segmentation of any test image I based on FCAMs follows the two successive stages outlined in Figure 5. The ELM is first trained with a set of $K_{I}\cdot T$ training samples, where *T* is the number of prototype sites per class that are randomly selected on each of the $K_{I}$ training images. Each training sample is the FCAM feature computed over a square patch W^*t*^, t∈{1,⋯,T}, centered at a prototype site and of size ${\left(2W+1\right)}^{2}$. The trained ELM is then used to segment I as follows. First, I is segmented into superpixels using the regional SLIC method. Then, the FCAM of each superpixel is fed as input into the trained ELM, whose output provides the estimated texture class. All sites in the superpixel are finally assigned to this class. A refinement procedure can be applied to further improve segmentation accuracy. This consists of reassigning superpixels smaller than 0.5% of the image size with the label of the largest adjacent superpixel [21]. All steps of the proposed color texture image segmentation are summarized in Algorithm 2.
**Algorithm 2:**Color texture image segmentation.**Input:** Test image I, $K_{I}$ training images
**Parameters:** Number *T* of prototypes per class, number *P* of superpixels, number *C* of fuzzy colors, patch half width *W*, membership function $\mu_{\tilde{c}}$.
**Step 1: Training stage**1.In each training image, randomly select *T* prototype sites.2.At each prototype site, compute the normalized FCAM ${\bar{m}}^{t}$ over a square patch W^*t*^ of size ${\left(2W+1\right)}^{2}$ using Equation (Equation 18).3.Train the ELM classifier with the $K_{I}\cdot T$ normalized FCAMs of prototypes.**Step 2: Segmentation stage**1.Run regional SLIC (Algorithm 1) on I to provide *P* superpixels ${\left\{P^{p}\right\}}_{p=1}^{P}$.2.Compute the normalized FCAM ${\bar{m}}^{p}$ of each superpixel $P^{p}$ using Equation (Equation 18).3.Feed ${\bar{m}}^{p}$ into the trained ELM and assign each pixel in $P^{p}$ to the ELM output class.**Step 3: Refinement (optional)****Output:** Segmented image

#### 4.1.3. Regional SLIC—Preliminary Assessment

To demonstrate the relevance of the proposed regional SLIC algorithm, we compare its results with those achieved by the basic SLIC algorithm thanks to four standard metrics. The achievable segmentation accuracy (ASA) quantifies the segmentation performance achievable by assigning each superpixel to the ground truth region with the highest overlap [55,59]. The boundary recall (BR) assesses the boundary adherence with respect to the ground truth boundaries [60]. The under-segmentation error (UE) evaluates the segmentation boundary accuracy as the overlap between superpixels and ground truth regions [55,59]. The compactness (COM) measures the compactness of superpixels [60]. Higher BR, ASA, and COM, and smaller UE, indicate better pre-segmentation.

Both the basic and regional SLIC algorithms require one to set the number of superpixels. This number must be small enough to get superpixels that are large enough to contain homogeneous textures, and small enough for the superpixels to finely fit boundaries between textures. We empirically found that P=400 is a good trade-off for the Prague dataset. Table 1 shows the average and standard deviation of the four metrics obtained by the basic and modified SLIC algorithms over the 20 images of Prague dataset. From this table, we can see that the results achieved by the regional SLIC are better than those obtained by the basic SLIC according to all metrics.

#### 4.1.4. Parameter Settings

The proposed color texture segmentation requires one to set several parameters. For the training stage, the number of prototypes was set to T=1000, since CNN-based methods use 1000 training texture mosaic images for each image to segment [23]. The size ${\left(2W+1\right)}^{2}$ of the patch centered at each prototype site and used for FCAM computation during the training stage was set according to the image size *N* and the number *P* of superpixels such that ${\left(2W+1\right)}^{2}\approx N/P$. As N=512×512 and P=400, the patch size was set to 27×27 pixels.

To characterize a site by an FCAM in the segmentation stage, we have to set both the number *C* of fuzzy colors and the membership function $\mu_{\tilde{c}}$ that defines the membership degree $\mu_{\tilde{c}}\left(s\right)$ of each site s to any fuzzy color $\tilde{c}$ according to its color I(s). We consider very few colors in order to evaluate how the memory cost of FCAMs can be reduced while preserving their relevance as texture features. Specifically, the number of fuzzy colors one of these values each time: C=2×2×1, C=2×2×2, C=2×3×2, C=2×4×2, C=3×3×2, C=3×4×2, or C=4×4×2, so that the FCAM size was 4×4, 8×8, 12×12, 16×16, 18×18, 24×24, or 32×32. Note that we privileged the *G* color component over *R* and *B* because it is similar to luminance; other combinations provide close classification accuracy results. Regarding the membership function, we had to choose one among the four functions (crisp, Gaussian, triangular, or FCM) presented in Section 3.4.

We retained a given number of fuzzy colors and membership function on the grounds of segmentation accuracy. To make this choice independent of the classification algorithm, we used the nearest neighbor classifier (1NN) instead of the ELM, and with no refinement step. Figure 6 shows the average accuracy obtained with each membership function according to the number *C* of fuzzy colors over the 20 images. From this figure, we can see that the fuzzy membership functions (Gaussian, triangular, and FCM) largely outperform the crisp one, especially when the number of fuzzy colors is low. However, when C≥12, all fuzzy membership functions lead to similar segmentation accuracies, even if the Gaussian membership function seems to perform slightly better. Moreover, accuracies do not vary significantly beyond C=16. In the following, all experiments were therefore performed with C=16 fuzzy colors and the Gaussian membership function.

### 4.2. Comparison with Other Fuzzy Texture Features

We here compare the relevance of FCAMs with respect to other fuzzy texture features, namely, the fuzzy gray-level co-occurrence matrix (FGLCM) [33], the fuzzy color co-occurrence matrix (FCCM) [38], and the fuzzy gray-level aura cardinal matrix (FGLAM) [39]. Note that FGLCM and FCCM are similar to a fuzzy color aura matrix computed with a fuzzy aura local measure, as shown in [39]. FGLCM and FGLAM features are formed by the concatenation of the marginal features of the three *R*, *G*, and *B* components; and a single FCCM captures the interactions among neighboring pixel colors. For a fair comparison, all these matrices were computed with the same parameters as the FCAM (Gaussian membership function, $C^{R}=C^{G}=C^{B}=16$ for FGLCM and FGLAM, and C=16 for FCCM).

The comparison is based on the segmentation accuracy as the main performance measure and on the computational time required by each feature extraction method. Experiments were carried out using the 1NN classifier instead of the ELM, on a computer with an Intel Core i7 3.60 GHz CPU and 8 GB of RAM. The results over the 20 images of the Prague dataset are summarized in Table 2. They clearly show that FCAM is more relevant, three times faster to compute, and requires less memory than marginal features (FGLCM and FGLAM). FCAM is also more efficient and slightly faster to compute than FCCM.

### 4.3. Comparison with State-of-the-Art Supervised Segmentation Methods

In this section, we compare the results obtained by the proposed segmentation method with the results of several state-of-the-art segmentation methods, on the Prague dataset. The assessment of segmentation performance was based on the conventional measures provided on the Prague texture segmentation website [57], which include: (1) *region-based criteria:* correct segmentation (CS), over-segmentation (OS), under-segmentation (US), missed error (ME), and noise error (NE); (2) *pixel-wise based criteria:* omission error (O), commission error (C), class accuracy (CA), recall (CO), precision (CC), type I error (I.), type II error (II.), mean class accuracy estimate (EA), mapping score (MS), root mean square proportion estimation error (RM), and comparison index (CI); (3) *consistency-error criteria:* global consistency error (GCE) and local consistency error (LCE).

The methods involved in the comparison were: (1) the MRF algorithm based on a Markov random field pixel classification model [11], (2) the COF algorithm that uses the co-occurrence features and the 1NN classifier [57], (3) the Con-Col algorithm [57], (4) the supervised fully convolutional network for texture (FCNT) algorithm [23] without refinement or (5) with refinement, (6) the empirical-wavelet-transform-based fully convolutional network for texture (EWT-FCNT) [23] that combines the empirical wavelet transform with FCNT, (7) U-Net [24], (8) the deep visual model (DA) [25], (9) the pyramid scene parsing network (PSP-Net) [26], and (10) our proposed method without refinement (FCAM nr) or (11) with refinement (FCAM wr).

The segmentation results of EWT-FCNT, FCNT, MRF, COF, and Con-Col were taken from the Prague benchmark website [57]; those of U-Net, DA, and PSP-Net were taken from [23]. All these methods except MRF, COF, and Con-Col, involve a refinement step to improve performance. Regarding the number of prototypes, no information is available for MRF, COF, and Con-Col. In contrast, the CNN-based methods used a training set of 1000 texture mosaic images of size 512×512 pixels, specifically created from the original Prague dataset, for each image to segment.

From Table 3, we can see that our proposed method, with and even without refinement, outperforms the classical supervised methods, namely, MRF, COF, and Con-Col.In contrast, EWT-FCNT and other CNN-based methods provided better segmentation results than our method. However, by carefully analyzing these results, we found that—except the EWT-FCNT method, which provided exceptional results—our method provided similar results (to U-Net, FCNT, DA, and PSP-Net). For example, the gaps between its accuracy (CO = 95.20% with refinement) and those of U-Net, FCNT, DA, and PSP-Net were lower than 1.7%. We can also see that our FCAM-based method did not suffer from under-segmentation nor over-segmentation (US and OS measures are equal to 0) compared to deep learning methods. It is also noticeable that our method without refinement outperformed FCNT without refinement according to most of the criteria.

To thoroughly analyze segmentation results, Table 4 presents the accuracy obtained on each individual test image by our proposed FCAM method; the handcrafted supervised methods MRF, COF, and Con-Col; and the two CNN-based methods, FCNT (with refinement) and EWT-FCNT. Only the segmentation results of individual test images obtained by the compared methods (EWT-FCNT, FCNT, MRF, COF, and Con-Col) are available on the Prague [57] benchmark website. Table 4 shows that accuracy obtained by our FCAM method was higher than 96% for half of the 20 test images and outperformed MRF, COF and Con-Col for almost all images. For 6 of the 20 tested images, our method also outperformed FCNT and ranked second behind EWT-FCNT. Even better, it provideed the best accuracy for image 11. For the other images, the accuracy obtained by our method is close to that of FCNT, though for images 7, 8, 10, and 13, the obtained accuracy is poor (below 91%). This explains why the average accuracy dropped to 95.20%.

Figure 7 shows a visual comparison of two handcrafted methods (COF and Con-Col) and two deep learning-based ones (FCNT wr and EWT-FCNT) with the proposed method. The results of these methods are publicly available online [57]. Overall, FCAM provided satisfactory visual segmentation results that are close to the ground truth. Unlike other methods, FCAM is not prone to over-segmentation, and classification errors almost only occurred at region boundaries. This is mainly due to our pre-segmentation based on regional SLIC that, despite its superiority over the basic SLIC, lacks the accuracy to correctly determine the boundaries between two or more different texture regions.

However, it is important to underline that deep learning-based methods (including EWT-FCNT) need to create a large training set (1000 images for each image to segment) from the original training database. Moreover, they require an expensive learning step to extract features and classify pixels. In order to get an overview on the computational costs of our approach compared to CNN-based methods, the following section gives the computational times measured during both training and segmentation phases.

### 4.4. Processing Time

In Section 4.2, we estimated the FCAM computational time only at superpixels. In order to provide an overview of the computational requirements of the entire FCAM-based color texture segmentation method, we estimated its overall runtime over the Prague dataset on an 3.60 GHz Intel Core i7 computer with 8 GB RAM. Table 5 displays the average computing time of our proposed method and those of CNN-based methods (EWT-FCNT, FCNT, U-Net, DA, and PSP-Net). These processing times are separated into two parts: the training time and the segmentation time. The segmentation times for CNNs were taken from [23]. They were measured on a laptop computer with 2.5 GHz quad-core Intel Core i7 processor and 16 GB memory, equipped with GTX 1080Ti external GPU with 11 GB memory. The training times of these CNNs were unfortunately not provided. From this table, we can see that the processing time required by our method during the segmentation stage is the highest. It is about 41.14 s for a 512×512 image. FCAM computation for superpixels consumes most of this time (33.12 s). The remaining time is mainly shared among the regional SLIC computation (8 s) and the classification of superpixels by ELM (0.02 s). It should be noted that the reported computing time for our method was obtained with MATLAB code without any GPU acceleration, and consequently, our method could be implemented for real-time applications with this enhancement.

In contrast, the total duration of the training stage for our method remained lower than 5 min and can be detailed as 260 s to compute the FCAM at the prototypes, and 0.9 s to train the ELM. The overall training time (260.9 s) was much lower than with CNNs, which typically require several hours on powerful computers equipped with large-memory GPUs.

## 5. Conclusions

In this paper, we introduced fuzzy color aura cardinal matrices (FCAMs) to locally characterize colors and textures, and applied them for color texture image segmentation. The FCAM feature makes it possible to locally characterize the interactions between colors of neighboring sites. A single low-dimensionnal FCAM is required to describe the color texture at each site, unlike in the marginal approach where a fuzzy gray-level aura matrix (FGLAM) must be computed for each color channel.

The proposed color texture image segmentation is based on the classification of superpixels, generated from a modified version of the SLIC algorithm to incorporate regional information. An FCAM is then computed for each superpixel thanks to a locally-adaptive neighborhood function. The superpixels are finally classified using a simple supervised ELM classifier. Experiments on the Prague texture segmentation benchmark showed that the proposed color texture segmentation based on FCAMs outperforms the classical state-of-the-art segmentation methods and is competitive with recent methods based on deep learning. However, unlike CNN-based approaches that require an expensive learning procedure and a large training set of segmented texture images, whose construction is time-consuming, our method is applied straightforwardly from a much smaller database using ELM-based classification.

Despite its respectable performance, our segmentation method remains sensitive to pre-segmentation results. Although the regional SLIC improves segmentation results in comparison with the basic SLIC, the detection of color texture boundaries is still not accurate enough. In future work, we intend to use a texture-aware superpixel procedure that takes into account the properties of the available color spaces. The proposed segmentation method was developed in a supervised context; we also plan to adapt it to unsupervised color texture segmentation.

## Figures and Tables

**Figure 1 jimaging-08-00244-f001:**
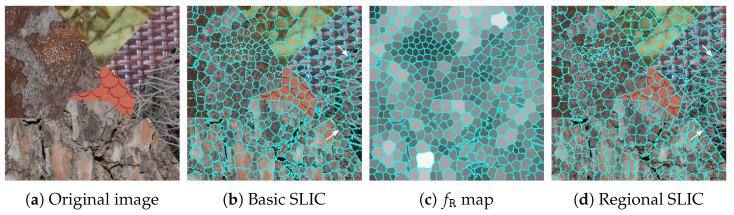
SLIC pre-segmentation results (P=400 superpixels). $f_{R}$ in (**c**) is rescaled to [[0,255]]. SLIC: simple linear iterative clustering.

**Figure 2 jimaging-08-00244-f002:**
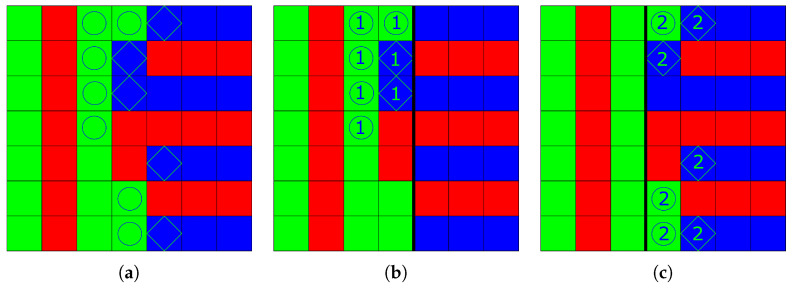
Examples of two aura sets in a color image. (**a**) Original image composed of 3 color site sets, $S_{R},S_{G}$, and $S_{B}$, and aura sets $A_{S_{G}}\left(S_{B}\right)$ (circles) and $A_{S_{B}}\left(S_{G}\right)$ (diamonds). (**b**) Superpixels $P^{1}$ (**left**) and $P^{2}$ (**right**), and aura sets $A_{S_{G}}^{1}\left(S_{B}\right)$ (①) and $A_{S_{B}}^{1}\left(S_{G}\right)$ (
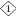
). $A_{S_{G}}^{2}\left(S_{B}\right)$ and $A_{S_{B}}^{2}\left(S_{G}\right)$ are empty. (**c**) Superpixels $P^{1}$ (**left**) and $P^{2}$ (**right**), and aura sets $A_{S_{G}}^{2}\left(S_{B}\right)$ (②) and $A_{S_{B}}^{2}\left(S_{G}\right)$ (
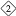
). $A_{S_{G}}^{1}\left(S_{B}\right)$ and $A_{S_{B}}^{1}\left(S_{G}\right)$ are empty.

**Figure 3 jimaging-08-00244-f003:**
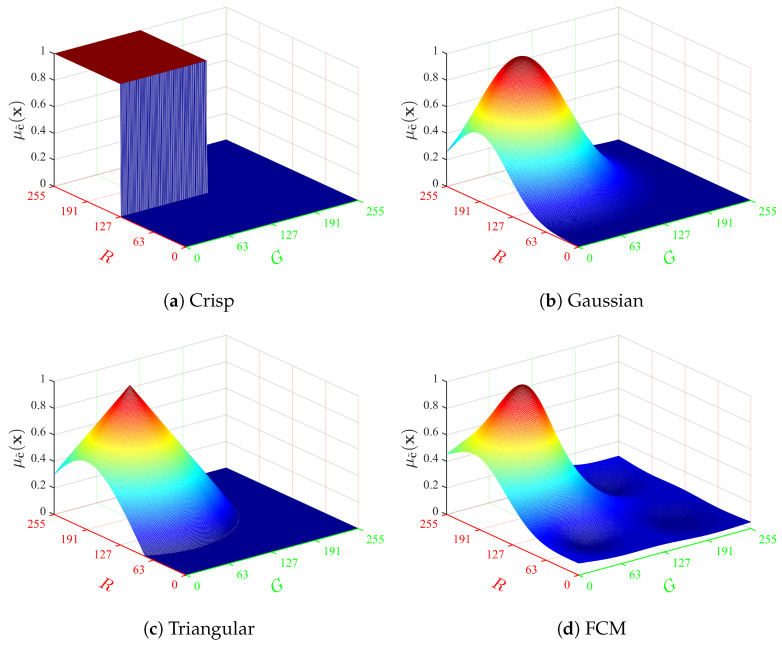
Membership functions $\mu_{\tilde{c}}$ at c=(192,64,64) with $C^{R}=C^{G}=C^{B}=2$.

**Figure 4 jimaging-08-00244-f004:**
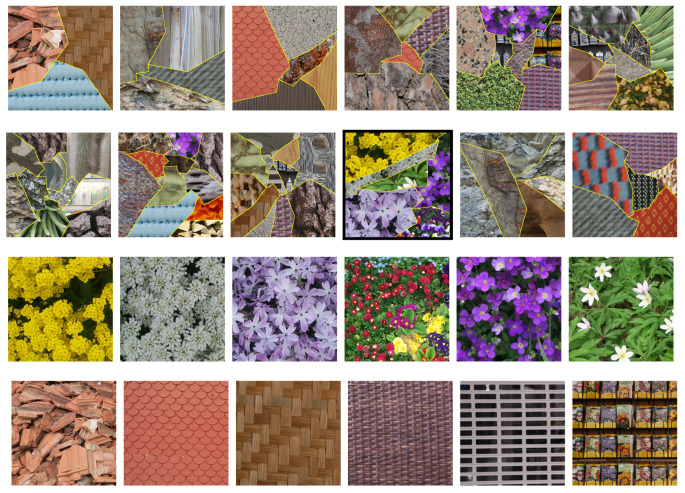
Examples of test images (**top** two rows) and training images (from “flowers” and “man-made” categories, **bottom** two rows) from the Prague dataset. Note that the framed test image represents $K_{I}=6$ classes from the sole “flowers” category.

**Figure 5 jimaging-08-00244-f005:**
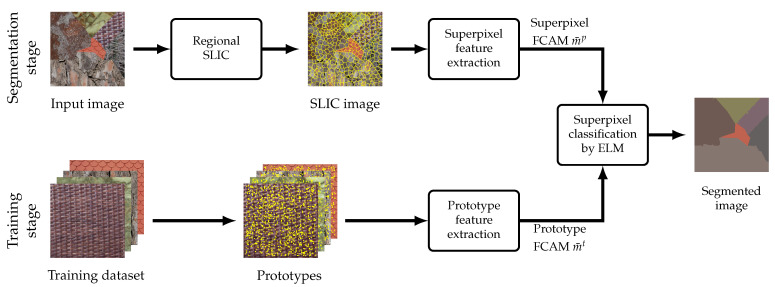
Flowchart of the proposed color texture image segmentation (SLIC: simple linear iterative clustering, FCAM: fuzzy color aura maxtrix, ELM: extreme learning machine).

**Figure 6 jimaging-08-00244-f006:**
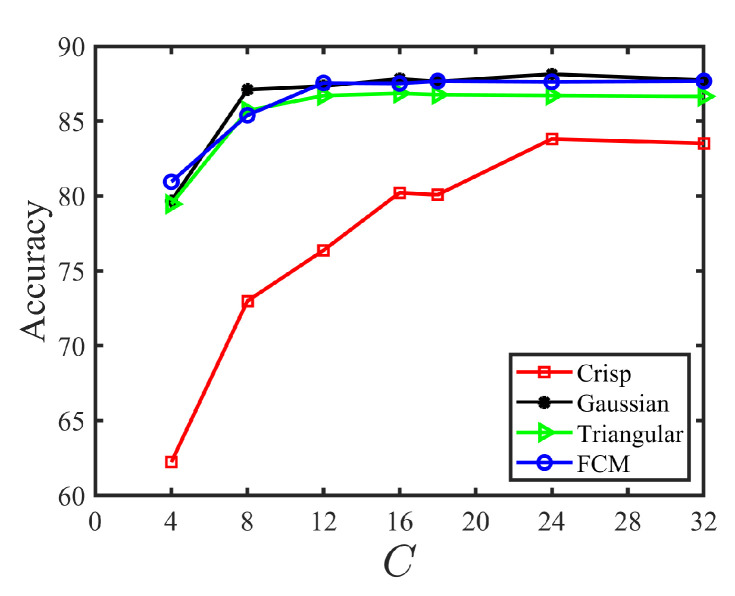
Average accuracy (%) vs. number of colors (*C*) for different membership functions.

**Figure 7 jimaging-08-00244-f007:**
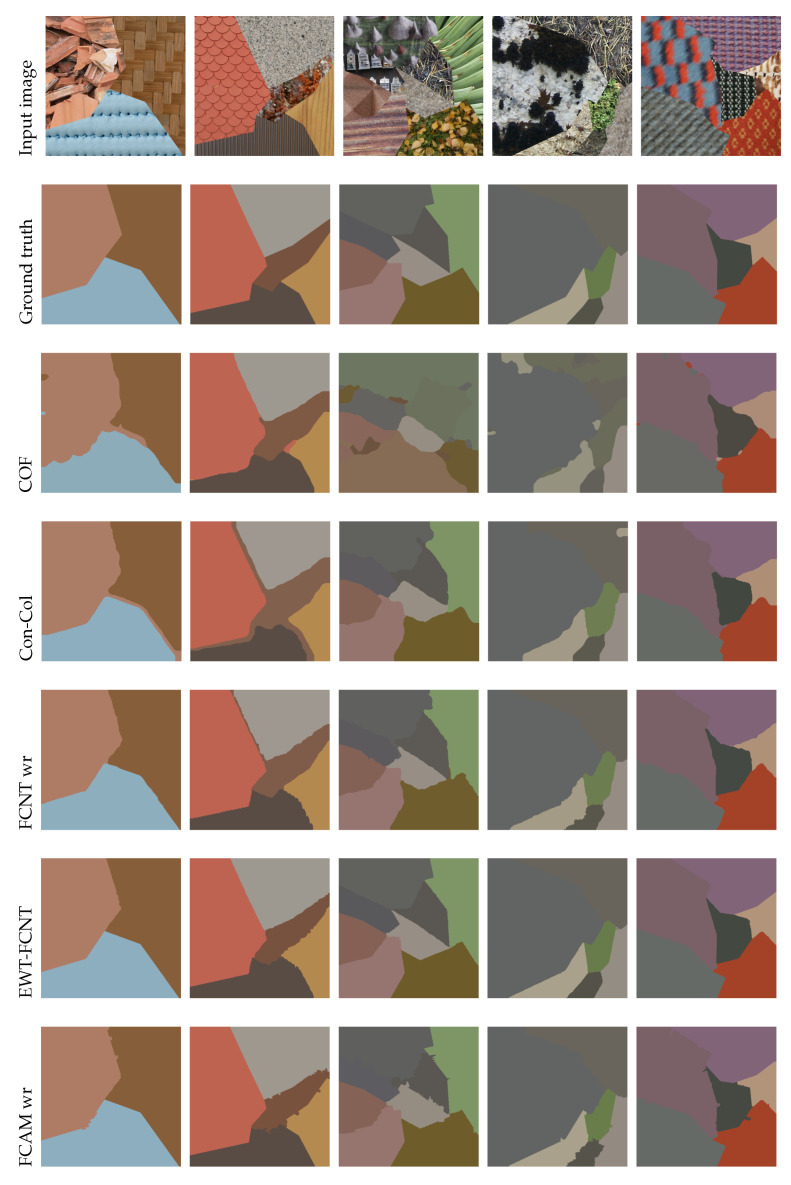
Sample segmentation results on Prague dataset, from left to right: image 01, image 03, image 06, image 15, and image 19.

**Table 1 jimaging-08-00244-t001:** Basic and regional SLIC performances on the Prague dataset. Arrows denote the metric direction (the higher ↑ or smaller ↓, the better), and best results are in bold.

Metric	↑BR	↓UE	↑ASA	↑COM
**Basic SLIC**	0.46±0.15	0.36±0.08	0.79±0.05	0.56±0.05
**Regional SLIC**	0.52±0.13	0.32±0.08	0.82±0.05	0.63±0.04

**Table 2 jimaging-08-00244-t002:** Average accuracy (%) and computation time (s) for different fuzzy features on the Prague dataset. “nr” means no segmentation refinement, and “wr” means with segmentation refinement.

Feature	Size	Accuracy nr	Accuracy wr	Comp. Time
FGLCM	$3\times 16^{2}=768$	86.69	90.97	123.53
FCCM	$16^{2}=256$	86.52	90.77	37.62
FGLAM	$3\times 16^{2}=768$	87.24	91.28	112.32
FCAM	$16^{2}=256$	**87.82**	**92.00**	**33.12**

**Table 3 jimaging-08-00244-t003:** Results of supervised methods on the Prague dataset (20 test images). Arrows ↑,↓ denote the required criterion direction; “nr” means no segmentation refinement, and “wr” means with segmentation refinement. Best results are in bold, and the second best ones in italic.

Criteria	MRF	COF	Con-	FCNT	FCNT	EWT-	U-Net	DA	PSP-	FCAM	FCAM
			Col	nr	wr	FCNT			Net	nr	wr
↑ CS	46.11	52.48	84.57	87.52	96.01	**98.45**	*96.71*	94.18	96.45	84.24	91.27
↓ OS	0.81	**0.00**	**0.00**	**0.00**	1.56	**0.00**	1.71	**0.00**	*0.17*	**0.00**	**0.00**
↓ US	4.18	1.94	1.70	**0.00**	1.20	**0.00**	**0.00**	1.18	*0.41*	**0.00**	**0.00**
↓ ME	44.82	41.55	9.50	6.70	0.78	**0.37**	*0.68*	3.42	1.23	11.45	5.93
↓ NE	45.29	40.97	10.22	6.90	0.89	**0.46**	*0.48*	3.24	1.12	11.39	5.36
↓ O	14.52	20.74	7.00	7.46	2.72	*0.93*	**0.72**	3.13	2.75	4.91	2.96
↓ C	16.77	22.10	5.34	6.16	2.29	*1.04*	**0.70**	1.32	2.39	5.89	2.72
↑ CA	65.42	67.01	86.21	87.08	93.95	**97.67**	*95.86*	94.53	93.89	87.28	91.54
↑ CO	76.19	77.86	92.02	92.61	96.73	**98.78**	*96.91*	96.23	96.06	92.60	95.20
↑ CC	80.30	78.34	92.68	93.26	97.02	**98.81**	*97.38*	97.01	96.41	93.65	95.96
↓ I.	23.81	22.14	7.98	7.39	3.27	**1.22**	*3.09*	3.77	3.94	7.40	4.80
↓ II.	4.82	4.40	1.70	1.49	0.68	**0.25**	*0.41*	0.58	0.69	1.32	0.87
↑ EA	75.40	76.21	91.72	92.68	96.68	**98.77**	*97.01*	96.24	96.08	92.58	95.13
↑ MS	64.29	66.79	88.03	88.92	95.10	**98.17**	*95.37*	94.35	94.08	88.90	92.80
↓ RM	6.43	4.47	2.08	1.38	0.86	**0.24**	*0.61*	1.07	0.70	1.64	1.24
↑ CI	76.69	77.05	92.02	92.81	96.77	**98.78**	*97.08*	96.41	96.15	92.84	95.35
↓ GCE	25.79	23.94	11.76	12.54	5.55	*2.33*	**2.13**	3.50	4.67	11.60	7.45
↓ LCE	20.68	19.69	8.61	9.94	3.75	*1.68*	**1.46**	2.47	3.52	8.76	5.31

**Table 4 jimaging-08-00244-t004:** Accuracy (CO) for each test image of the Prague dataset. “nr” means no segmentation refinement, and “wr” means with segmentation refinement. Best results are in bold, and the second best ones in italic.

Image	MRF	COF	Con-	FCNT	EWT-	FCAM	FCAM
			Col	wr	FCNT	nr	wr
01	*99.79*	96.19	96.92	99.12	**99.91**	99.54	99.72
02	77.36	93.02	92.70	*97.71*	**99.51**	92.66	96.79
03	95.60	96.56	92.33	97.47	**99.21**	98.66	*99.03*
04	73.20	68.63	93.73	98.41	**98.77**	96.38	*98.58*
05	89.72	89.67	91.64	*96.64*	**98.95**	92.95	93.83
06	84.00	57.59	95.78	97.30	**99.52**	96.97	*97.64*
07	67.74	58.08	89.71	*96.09*	**96.71**	79.65	84.70
08	58.12	71.27	90.10	*95.67*	**98.80**	84.16	90.21
09	72.95	61.29	93.23	*96.70*	**99.35**	94.90	95.47
10	71.52	58.97	85.72	*92.52*	**96.69**	86.36	89.51
11	61.28	80.88	*96.82*	96.72	95.32	94.44	**97.12**
12	81.55	63.01	87.20	*96.03*	**99.51**	92.74	95.84
13	74.35	84.32	77.02	*96.10*	**98.48**	87.99	90.11
14	91.29	90.32	96.51	*97.75*	**99.17**	94.25	96.39
15	57.77	79.61	96.26	97.70	**99.56**	95.31	*98.61*
16	61.33	60.63	91.19	*94.31*	**99.46**	86.23	93.92
17	62.74	72.31	91.92	*96.96*	**99.38**	91.21	95.10
18	77.81	92.96	96.16	98.24	**99.58**	97.18	*98.54*
19	76.07	94.98	97.02	98.55	**99.78**	97.64	*99.39*
20	89.68	86.87	88.48	*94.66*	**97.91**	92.82	93.44
Average	76.19	77.86	92.02	*96.73*	**98.78**	92.60	95.20

**Table 5 jimaging-08-00244-t005:** Average computation time per image of FCNT, EWT-FCNT, U-Net, DA, PSP-Net, and FCAM on the Prague dataset.

Stage	FCNT	EWT-FCNT	U-Net	DA	PSP-Net	FCAM
**Segmentation**	3.61 ms	1.830 s	4.98 ms	3.80 ms	14.39 ms	41.14 s
**Training**	-	-	-	-	-	260.9 s

## Data Availability

Not applicable.

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
