# Peer review of "Fuzzy Color Aura Matrices for Texture Image Segmentation"

_2313-433X, 2022, doi:10.3390/jimaging8090244_

Round 1
Reviewer 1 Report
The proposed approach has some novelty in contribution. Revision in terms of technical details is needed. So, some comments are suggested to describe technical details.
1. It is suggested to discuss about the runtime of your proposed approach. (new experiments are not needed) Is it fast enough to perform your proposed approach in real applications?
2. How do you tune the square patch size in your experiments? Is it selected randomly?
3. Is it possible to extend your proposed approach to different color spaces? Some color spaces have less than 3 sites!
4. Color texture classification can be used in medical applications as one of the important usability scopes. I find a paper titled “Detection of circulating NK cells in Breast Cancer Patients Under Chemotherapy in Al-Hillah City, Iraq”, which has enough relation. Cite this paper and discuss about applications briefly.
5. Did you evaluate the performance of your proposed approach in terms of different disjoint intervals?
6. Since now, many different approaches has been proposed for color texture segmentation and color texture classification. S, it is suggested to discuss about related works briefly in the section 1.
6. Did you implement all of the compared methods in the Table 3? If no, add related references
Author Response
Dear Reviewer 1,
Please read the PDF file.
Best,

Reviewer 2 Report
1. Before the spelling out, do not use the abbreviation such as SLIC in the abstract. In addition, for the smoothing reading, try to avoid any abbreviations in the abstract.
2. Authors claim that "Experimental results on the Prague texture segmentation benchmark show that our method outperforms the classical state-of-the-art supervised segmentation methods and is competitive with recent methods based on deep learning", however, this claim does not match well with the results shown on Table 3.
3. The authors tested on 20 images, what is the best result and worst result for all the images tested by each method? I assume that Table 3 shows the average result. The reader might be interested in seeing the overall of these, not just the average result.
4. The authors used too many different measures for the segmentation performance assessment. It is not necessary to use them all as many of them are similar.
5. Figure 6 shows the accuracy with different membership functions. This is not very critical for authors to show those functions and results.
Author Response
Dear Reviewer 2,
Please read the PDF file.
Best,

Round 2
Reviewer 1 Report
The revised version is better than original submission in terms of technical details. Add descriptions improve the quality of presentation. It is suggested to explain in what directions the operator is calculated.